# Medium-Term Nutritional and Metabolic Outcome of Single Anastomosis Duodeno-Ileal Bypass with Sleeve Gastrectomy (SADI-S)

**DOI:** 10.3390/nu15030742

**Published:** 2023-02-01

**Authors:** Giuseppe Marincola, Valeria Velluti, Nikolaos Voloudakis, Pierpaolo Gallucci, Luigi Ciccoritti, Francesco Greco, Luca Sessa, Giulia Salvi, Amerigo Iaconelli, Barbara Aquilanti, Caterina Guidone, Esmeralda Capristo, Geltrude Mingrone, Francesco Pennestrì, Marco Raffaelli

**Affiliations:** 1U.O.C. Chirurgia Endocrina e Metabolica, Centro Dipartimentale di Chirurgia Endocrina e dell’Obesità, Fondazione Policlinico Universitario Agostino Gemelli IRCCS, 00168 Roma, Italy; 2U.O.S.D. Medicina Bariatrica, Fondazione Policlinico Universitario Agostino Gemelli IRCCS, 00168 Roma, Italy; 3Dipartimento di Medicina e Chirurgia Traslazionale, Università Cattolica del Sacro Cuore, 00168 Roma, Italy; 4Centro di Ricerca in Chirurgia delle Ghiandole Endocrine e dell’Obesità, Università Cattolica del Sacro Cuore, 00168 Roma, Italy; 5Centro Malattie Endocrine e Obesità, Fondazione Gemelli Giglio Cefalù, 90015 Cefalù, Italy; 6U.O.C. Patologie dell’Obesità, Fondazione Policlinico Universitario Agostino Gemelli IRCCS, 00168 Roma, Italy

**Keywords:** SADI-S, nutritional deficiency, hypoabsorptive surgery, metabolic results

## Abstract

Introduction: Single Anastomosis Duodenal-Ileal Bypass with Sleeve Gastrectomy (SADI-S), like other hypoabsorptive procedures, could be burdened by long-term nutritional deficiencies such as malnutrition, anemia, hypocalcemia, and hyperparathyroidism. Objectives: We aimed to report our experience in terms of mid-term (2 years) bariatric, nutritional, and metabolic results in patients who underwent SADI-S both as a primary or revisional procedure. Methods: One hundred twenty-one patients were scheduled for SADI-S as a primary or revisional procedure from July 2016 to February 2020 and completed at least 2 years of follow-up. Demographic features, bariatric, nutritional, and metabolic results were analyzed during a stepped follow-up at 3 months, 6 months, 1 year and 2 years. Results: Sixty-six patients (47 female and 19 male) were included. The median preoperative BMI was 53 (48–58) kg/m^2^. Comorbidities were reported in 48 (72.7%) patients. At 2 years, patients had a median BMI of 27 (27–31) kg/m^2^ (*p* < 0.001) with a median %EWL of 85.3% (72.1–96.1), a TWL of 75 (49–100) kg, and a %TWL of 50.9% (40.7–56.9). The complete remission rate was 87.5% for type 2 diabetes mellitus, 83.3% for obstructive sleep apnea syndrome and 64.5% for hypertension. The main nutritional deficiencies post SADI-S were vitamin D (31.82%) and folic acid deficiencies (9.09%). Conclusion: SADI-S could be considered as an efficient and safe procedure with regard to nutritional status, at least in mid-term (2 years) results. It represents a promising bariatric procedure because of the excellent metabolic and bariatric outcomes with acceptable nutritional deficiency rates. Nevertheless, larger studies with longer follow-ups are necessary to draw definitive conclusions.

## 1. Introduction

Single-Anastomosis Duodeno-Ileal Bypass with Sleeve Gastrectomy (SADI-S) is a hypoabsorptive bariatric procedure first described by A. Torres and co-workers in 2007 as a modification of the standard Biliopancreatic Diversion with Duodenal Switch (BPD-DS) to simplify the surgical technique and reduce the nutritional deficits [1]. SADI-S can be recommended as a primary procedure for complex bariatric patients (Body Mass Index-BMI > 50 kg/m^2^) and/or for metabolic patients (with comorbidities related to obesity, especially type 2 diabetes mellitus—T2DM) [2]. It may also be recommended as revisional surgery in patients who failed previous bariatric procedures, e.g., after sleeve gastrectomy [3] (Single-Anastomosis Duodeno-Ileal Bypass-SADI) [4]. SADI-S may also be considered as a second operation in patients who failed Adjustable Gastric Banding (AGB) or Roux-en-Y Gastric Bypass (BPG) [5].

The SADI-S procedure implies the preservation of the antropyloric region using an omega loop reconstruction with an afferent and absorbent intestinal limb [6]. This feature has important physio-pathological implications and can improve the assimilation of micro- and macronutrients, especially group B vitamins, reducing the possible metabolic complications thanks to the maintenance of pyloric function. In the original description, the first 50 patients underwent this bariatric procedure with a common limb of only 200 cm. Despite the good bariatric results, there was a high rate of malnutrition (8%) [7]. For this reason, the common limb was extended to 250 cm or 300 cm, which are the standardized current measures, with the latter representing the best cost-benefit ratio according to many authors [8].

SADI-S is a safe and validated procedure with important bariatric and metabolic outcomes and an acceptable postoperative complication rate, as reported in the latest IFSO Position Statement 2020 [7].

However, recent meta-analyses showed that, although it is a safe and efficient procedure, the range of results obtained is very wide [9,10].

However, the mid-term (up to 5 years) nutritional outcome following SADI-S has not been extensively evaluated yet, and the pertinent literature has not yielded definitive results. To date, no “gold standard treatments” are established, and definitive nutritional supplementation data are currently lacking.

The aim of our retrospective study is to evaluate nutritional, bariatric, and metabolic outcomes at a mid-term follow-up (2 years after surgery) in patients who underwent SADI-S in our center, both as a primary and as a revisional procedure.

## 2. Methods

From July 2016 to February 2020, 2313 bariatric procedures were performed (2078 primary procedures and 235 revisional procedures). A total of 121 patients were scheduled for SADI-S/SADI. Among them, 66 patients completed the nutritional follow-up at 2 years following the procedure and were included in the present study. The follow up for this study ended on 28 February 2022.

Patients included in this study met the consensus criteria for bariatric surgery, fulfilled the national guidelines of Italian Society of Bariatric Surgery and Metabolic Disorders (SICOB) [https://www.sicob.org/00_materiali/linee_guida_2016.pdf (accessed on 20 December 2022)], and underwent the primary (SADI-S) or revisional procedure (SADI) either with a laparoscopic or robotic approach. Patients were fully informed of the surgical technique, anesthesia, effects, and complications. 

The preoperative workup consisted of an upper endoscopy, ultrasound of the abdomen, upper gastrointestinal (UGI) contrast study, blood analysis, respiratory investigation, nutritional status appraisal, and psychological and cardiac evaluations. Multidisciplinary evaluation (by a team consisting of a surgeon, an endocrinologist, a dietician, and a psychologist) was performed for every patient with the aim to have a personalized bariatric process reported in detail [11].

The description of the surgical procedures has already been reported [10,12].

Primary endpoint: assessment of nutritional status at midterm follow-up (2 years) in patients who underwent SADI-S/SADI.

Secondary endpoints: Assessment of perioperative complications (<30 days after surgery) and late complications after surgery, metabolic and bariatric outcomes in patients who underwent SADI-S/SADI.

### 2.1. Post-Operative Protocol

A standardized postoperative protocol tailored to bariatric patients was used [10,12]. The severity of postoperative complications was graded according to the Clavien-Dindo classification [13]. A routine follow-up with blood test analysis and physical examination was performed according to the guidelines of the Italian Society of Bariatric Surgery (SICOB) [https://www.sicob.org/00_materiali/linee_guida_2016.pdf, (accessed on 20 December 2022)]. At discharge, patients were advised to follow a strict diet consisting of three progressive phases (clear liquid, semi-solid, and solid), each lasting at least 2–3 weeks and supplemented with proteins, vitamins, and minerals (Table 1). Protein supplementation was indicated because clinical practice guidelines for perioperative support of bariatric patients by the Bariatric Surgery Societies (SICOB, IFSO) recommend a daily protein intake of at least 60 up to 1.5 g/kg for an ideal body weight. All patients received directions to buy a particular vitamin and mineral bariatric supplement (FitForMe WLS Maximum^®^, Fit for Me, DA Rotterdam, The Netherlands) that is tailor-made for bariatric patients who have undergone hypoabsorptive procedures (see Table 2 for composition). Patients purchased the product at their own expense. 

Vitamin supplementation during the study period was highly recommended. All patients received enoxaparin (4000 UI/0.4 mL) for 4 weeks and a proton pump inhibitor (PPI) (esomeprazole, 40 mg daily) for at least 6 months as part of the standard postoperative protocol.

### 2.2. Definitions

Baseline demographic and clinical data (baseline (T0), intraoperative, postoperative, and 24 months (T1)) were collected by reviewing patient records and electronic databases: age, BMI, gender, comorbidities, hemoglobin (reference range 12.0–15.0 g/dL in female patients and 13.0–17.0 g/dL in male patients), total protein (reference range 65–85 g/L), albumin (reference range 34–48 g/L), potassium (reference range 3.0–5.0 mmol/L), sodium (reference range 135–145 mmol/L), chloride (reference range 98–108 mmol/L), high-density lipoprotein cholesterol (HDL) (reference cut-off >40 mg), low-density lipoprotein cholesterol (LDL) (reference cut-off 130 mg/dL), glycemia (reference range 65–110 mg/dL), HbA1c (reference range 23.0–41.0 mmol/mol), parathyroid hormone (PTH) (reference range 14–72 pg/mL),serum folic acid (reference cut-off >4 ng/mL), vitamin B12 (reference range 187–883 pg/mL), vitamin D (reference range 31–100 ng/mL). Calcium levels were corrected for albumin using the following equation: corrected calcium = measured total calcium (mg/dL) + 0.8 × (4-serum albumin (g/dL)), where 4 represents the average albumin level. All biochemical tests were performed in the same laboratory of our center to avoid possible bias.

Other parameters such as postoperative pain (Visual Analogue Scale—VAS scale, 0–10), nausea, vomiting, drain output, urine output, hemoglobin level, leukocyte level, need for blood transfusion, and surgical findings (if further surgery was required) were also recorded. 

Preoperative vitamin deficiencies were treated with a specific integration, and effective correction was verified with blood tests before surgery to minimize selection bias. After surgery, further multivitamin and trace element supplements were taken to correct specific deficiencies.

The percent loss of excess body weight (%EWL) was calculated as ((baseline weight − postoperative weight)/(baseline weight − ideal body weight)) × 100. Ideal body weight was calculated using the weight equivalent to a BMI of 25 kg/m^2^. The percent loss of total body weight (%TWL) was calculated as ((baseline weight − postoperative weight)/baseline weight)) × 100.

Total body weight loss (TWL) was calculated as (baseline weight − postoperative weight) expressed in kilograms. 

Anemia was defined as serum hemoglobin concentrations lower than 12 g/dL for females and lower than 13 g/dL for males. Hypoprotidemia was defined as serum total proteins concentrations lower than 65 g/L. Hypoalbuminemia was defined as serum albumin concentrations lower than 34 g/L. Hypocalcemia was defined as serum calcium concentrations lower than 8.6 mg/L. Hyposodemia was defined as serum sodium concentrations lower than 135 mmol/L. Hypokaliemia was defined as serum potassium concentrations lower than 3 mmol/L. Hypochloridemia was defined as serum chloride concentrations lower than 3 mmol/L. Hypovitaminosis D was defined as serum 25-hydroxy vitamin D concentrations lower than 31 ng/mL. Hypovitaminosis B12 was defined as serum vitamin B12 concentrations lower than 187 pg/mL. Hypovitaminosis B9 was defined as serum vitamin B9 concentrations lower than 4 ng/mL. Hyperparathyroidism was defined as serum PTH concentrations levels higher than 72 pg/mL. Hyperglycemia was defined as serum glucose concentrations levels higher than 110 pg/mL.

After SADI-S/SADI, partial or complete resolution of comorbidities was defined as the following: partial resolution was considered as a reduction in preoperative treatment, while complete resolution of comorbidities was defined as the discontinuation of any treatment and reversal to normality of altered laboratory parameters; diabetes resolution was considered partial when the glycated hemoglobin levels were <6.5%, and fasting blood sugar levels of 100–125 mg/dL were achieved. Complete remission is regarded as reaching a value of glycated hemoglobin of <6% and a value of fasting blood sugar of <100 mg/dL. For hypertension, partial and complete resolution was defined as a reduction in or suspension of antihypertensive therapy, respectively. Regarding OSAS, partial or complete remission is defined as an improvement or a normalization of the polysomnography, respectively, or a reduction in or suspension of CPAP therapy, respectively.

### 2.3. Statistical Analysis

Baseline demographic and clinical data were collected by reviewing patient records and electronic databases. Data from all patients scheduled for SADI-S were prospectively collected. Statistical analysis was performed using SPSS 22.0 software for Windows (SPSS Inc, Chicago, IL, USA). Continuous variables were expressed as the median (interquartile range, IQR). Dichotomous variables were expressed as the number and percentage. The presence of a normal distribution was tested using the Shapiro–Wilks test. Differences between baseline and the different time points of follow-up were analyzed with the repeated measures t-test or the non-parametric Friedman test depending on the distribution of the data in the analyzed population. The Cochran’s Q test was used for the comparison of categorical variables. We referred to a 5 percent significance level. 

This study was conducted according to the guidelines of the Declaration of Helsinki. The study was approved by our institution’s ethical committee.

## 3. Results

During the study period, 2313 bariatric procedures were performed (2078 primary procedures and 235 revisional procedures). A total of 121 patients were scheduled for SADI-S/SADI. In 66 patients, a complete nutritional and metabolic follow-up was performed 2 years after the procedure. 

Baseline demographic and clinical characteristics of the included patients are reported in Table 3.

The median age of the patients included in our study was 42 years (range: 38–50), and the preoperative BMI was 53 (range: 48–53) kg/m^2^; 19 patients (28.78%) were males and 47 (71.22%) females. 

The median time between the primary bariatric procedure and the revisional SADI for weight regain of inadequate weight loss was 48 months (range: 36–96) after sleeve gastrectomy and 126 (range: 96–156) after adjustable gastric banding. 

The median BMI before sleeve gastrectomy was 50 (range: 46–57) kg/m^2^, and the median BMI before adjustable gastric banding was even higher, 54 (range: 46–57) kg/m^2^.

A total of 18 patients (27.3%) had no comorbidities related to obesity while 42 (72.7%) had at least one comorbidity (hypertension, diabetes, and obstructive sleep apnea syndrome—OSAS); 30 (45.5%) patients had OSAS; 31 (47.0%) patients had hypertension; and 16 patients (24.2%) had T2DM. At the 2-year follow-up, two patients were deceased.

A 42-year-old female patient, with a preoperative BMI of 53.1 kg/m^2^, died two years after laparoscopic surgery because of sepsis following pneumonia. She was severely malnourished because she refused any supplementation and nutritional advice and was lost at follow-up, despite frequent calls by the nutritionist team. A 37-year-old male patient, with a preoperative BMI of 66.3 kg/m^2^, died 18 months after SADI-s because of polytrauma (car accident). He was in good health.

### Nutritional and Metabolic Outcome 

The anthropometric results are shown in Table 4.

After 2 years, the median BMI was significantly lower compared to the preoperative value (53, range: 48–58 kg/m^2^, vs. 27, range: 23–31 kg/m^2^, *p* < 0.001). The median total weight loss was 75 (range: 49–100) kg. At the median follow-up of 2 years, the %TWL was 50.9% (range: 40.7–56.9), and the %EWL was 85.3% (range: 72.1–96.0).

Nutritional outcomes and resolution of comorbidities are shown in Table 5 and Table 6.

In the preoperative period, 2 patients (3.00%) had folic acid deficiency; 8 patients (12.1%) had hypovitaminosis D; 1 patient (1.50%) had hypoalbuminemia; and 12 patients (18.1%) had anemia. None had vitamin B12 deficiency or hypoproteinemia (<50–55 g/L). All nutritional deficiencies were corrected before surgery. 

After SADI-S, 2 patients (3.03%) experienced anemia, 1 patient (1.52%) hypoproteinemia and 3 patients (4.52%) hypoalbuminemia.

No patient experiencing postoperative hypocalcemia, hyponatremia, hyperglicemia, or reactive hypoglicemia was observed. Two patients (3.03%) experienced slight, self-limiting hypokalemia. 

A total of 21 out of 66 patients (31.82%) experienced vitamin D deficiency after surgery, and 6 out of 66 patients (9.09%) experienced folic acid deficiency. 

OSAS completely resolved in 25 patients (83.33%); hypertension completely resolved in 20 patients (65.50%); and T2DM completely resolved in 14 patients (87.50%) of the initially affected patients.

## 4. Discussion

The results of the present study report mid-term outcomes of SADI-S performed at a high-volume bariatric referral center from July 2016 to February 2020. 

Concerning the primary outcome of the study, we found negligible micronutrients deficiencies, with a good nutritional status at mid-term (2 years) in patients who underwent SADI-S/SADI.

SADI-S is a relatively new surgical technique introduced by Sanchez-Pernaute and Torres et al. in 2007 in order to obtain similar results, such as excess weight loss, as BPD-DS but with lower rates of complications [1]. The safety of the procedure has been adequately demonstrated compared with BPD/DS [14]. Over time, the technique has been further improved. Initially, the common limb was 200 cm, and, although the %EWL was about 100% [15], this length caused severe malnutrition in patients. For this reason, in 2009, the length of the common limb was increased to 250–300 cm for most patients [15]. By performing a single anastomosis, both the rate of complications and surgical times are reduced as well as the overall anesthesia time.

Hypoabsorptive bariatric operations, including SADI-S, are associated with a high prevalence of nutritional deficiencies due to the altered gastrointestinal anatomy and reduced absorptive capacity. In the initial period after surgery, patients have a reduced food intake and dyspepsia. These conditions can lead to severe malnutrition related to reduced caloric intake or nutrient deficiency [16,17]. Considering the quite relevant percentage of obese patients with vitamin and micronutrient deficiencies even prior to surgery and their modest adherence with the oral supplementation, any hypoabsorptive procedure may be associated with catastrophic nutritional and metabolic outcomes. That underlines once more the importance of accurate selection and strict nutritional follow-up in patients undergoing SADI-S. However, for any bariatric procedure, at least a yearly nutritional follow-up is mandatory [17,18].

The yearly nutritional follow-up is highly recommended to ensure a personalized treatment [19,20]. Currently, there is no specific evidence about the treatment of micronutrients deficiencies in patients that underwent SADI-S, but existing guidelines suggest nutritional recommendations similar to BPD-DS treatment. Unfortunately, in the current literature, the number of studies with long-term follow-up on the nutritional status after SADI-S are limited and controversial [19].

Hypoabsorptive procedures are associated with higher fat malabsorption rates and may also cause significant fat-soluble vitamin deficiencies, more specifically vitamin D, while also affecting calcium levels [21,22]. In addition, total protein and albumin deficiency is a common condition in operated obese patients. Another common complication following bariatric surgery is iron deficiency along with microcytic anemia and can be partially attributed to reduced intestinal absorption and/or menstruation in young women. The presence of anemia can also be justified by the absence of an acidic environment and of the Intrinsic Factor, produced by the gastric parietal cells, which are required for Vitamin B12 and folic acid absorption [23]. On the other hand, outcomes of nutritional deficiencies described in the literature are more controversial and often conflicting. Reported SADI-S nutritional outcomes differ among various centers. Several factors might influence those variable results, such as patients’ selection, supplementation protocols applied, center’s expertise, and adherence to follow-up. This can be seen among some prominent publications; Balibrea et al. [24] presented the mid-term results after 30 SADI in both failed SG and two-step strategy in super obese patients, illustrating a clear relationship between common channel length and nutritional deficiencies and reporting a deficiency of total protein, folate, vitamin B12, vitamin D, and calcium in 58.33%, 18.18%, 33.33%, 55.56%, and 45.45% of patients, respectively. This global nutritional deficiency rate is possibly due to the lower supplementation (especially Vitamin D) prescribed compared with the recommended levels after BPD-DS. On the other hand, Moon et al. [25] reported two-year outcomes in 140 primary laparoscopic and robot-assisted SADI-S describing lower albumin levels in 20.0%, total protein in 12.0%, and calcium in 16% of patients. Sanchez-Pernaute et al. [26], concerning mid-term results of 97 patients with T2DM following SADI-S, reported a deficiency of total proteins, albumin, and vitamin D in 34%, 13.7%, and 50% of patients, respectively. However, no detailed information concerning postoperative supplementation was reported by the authors. Zaveri et al. [27], in a cohort of 286 patients, demonstrated lower levels of calcium, total protein, albumin, vitamin B12, and vitamin D in 4.89%, 5.59%, 3.46%, 0%, 15.03% of patients, respectively, after a 2-year follow-up. The differences among various published series can be partially attributed to different patients’ selection and management protocols adopted. Few publications include, or clearly elucidate, the nutritional protocols, supplementation details, and adherence to follow-up when reporting their results. In our study, the few cases that presented with nutritional deficiency can be justified by an inadequate adherence to prescribed therapy, diet, or follow-up. 

It should also be noted that in our clinical practice we use very strict cut-offs to define nutritional deficiencies. While this clearly allows an early identification and treatment of these conditions, it also results in slightly higher rates than those reported by many authors who instead use more tolerable ranges.

Current nutritional recommendations for malabsorptive bariatric procedures, such as BPD-DS, suggest that monitoring should take place at 3, 6, and 12 months in the first year and at least annually thereafter [23,28].

Concerning weight loss, the efficacy of SADI-S was previously demonstrated, and our results are in line with those reported in the pertinent literature. At 24 months following surgery [2,20,24], the reported BMI varied from 28.6 to 32.7 kg/m^2^, %EWL from 73.91% to 85.96%, and %TWL from 25.8% to 46.3% [27,29]. The differences in the published results may be attributed to a variable common loop length.

The efficacy of SADI-S on the obesity-related comorbidities’ resolution has been widely established. The complete resolution of T2DM was observed in 63.7% to 78.6% of patients following surgery [26,27]. For hypertension, complete resolution ranged between 42.4% and 66.4% [25,27,30,31], while for OSAS, resolution was observed in 47.4% to 59.6% of patients in large series [25,27,31].

We believe that a multidisciplinary approach, consisting of highly specialized professionals, including surgeons, endocrinologists, dieticians, and psychologists, and a regular long-term follow-up, is paramount in patients operated on with SADI-S. In our opinion, the prescription of a specific diet and a targeted therapy for hypoabsorptive bariatric procedures such as SADI-S is mandatory to prevent the onset of critical micronutrients deficiencies.

The role of the multidisciplinary team is fundamental also in the preoperative period for appropriate patient selection for surgery, taking into consideration not only clinical factors but also socio-economic and behavioral conditions in order to assure a correct adherence to the follow-up and therapy.

This condition is possible in high volume bariatric referral centers, with defined and organized follow-up programs, based on current guidelines, specific for bariatric patients.

To summarize, our results are extremely encouraging, with an excellent weight loss rate, a good control of comorbidities, and satisfactory nutritional outcomes. The complications that were observed, which often go beyond the type of intervention and the nutritional deficiencies, were often associated with poor adherence to therapy, diet, and scheduled follow-up or with concomitant pathologies.

Several limitations of our analysis should be noted. Firstly, this is a retrospective study over a long period of time. Secondly, the nutritional and metabolic results over a longer follow-up have not been analyzed yet since the number of patients completing the longer follow-up is still limited.

Our analysis was based on data from real-world clinical practice. Therefore, it is important to emphasize that the measurement of vitamin and trace element levels has a cost that is not completely supported by our national health system. For this reason, we were unable to analyze certain important nutrients due to lack of data.

It is evident that randomized control trials should be performed in the future in order to assess alternate follow-up regiments and specific therapies. In addition, further randomized studies comparing SADI-S/SADI with other bariatric procedures, including longer follow-up times and adopting a prospective nature, should be carried out to consolidate the potential benefits of SADI-S.

## 5. Conclusions

SADI-S is a safe and effective procedure with promising mid-term (2 years) bariatric and metabolic results and acceptable postoperative nutritional complications. It could represent an interesting option for the management of complex bariatric (metabolic and/or super-obesity) patients as it is burdened by lesser nutritional sequelae typically associated with the gold standard hypoabsorptive bariatric procedures. Nevertheless, SADI-S could also serve as an optimal choice of revisional surgery after a failed bariatric surgery. However, further studies with longer follow-up data are needed to draw definite conclusions.

## Figures and Tables

**Table 1 nutrients-15-00742-t001:** Diet Stages after Bariatric Surgery.

Diet	Kcal	Food/Liquid Intake	Dietary Composition
Liquid diet	300 Kcal	Clear liquids	Protein 47%, Lipids 6%, Carbohydrates 47%
Semisolid diet	750–800 Kcal	Blended, soft, and puréed foods	Protein 37%, Lipids 18%, Carbohydrates 45%
Solid diet	1000–1200 Kcal	Minced and solid foods	Protein 25%, Lipids 28%, Carbohydrates 47%

**Table 2 nutrients-15-00742-t002:** Composition of vitamin and mineral bariatric supplement.

Nutrient	Dose	RI *
Vitamin A	1200 µg RE	150%
Vitamin B1	3 mg	273%
Vitamin B2	3.5 mg	250%
Vitamin B3	32 mg NE	200%
Vitamin B5	18 mg	300%
Vitamin B6	1.4 mg	100%
Vitamin B8	100 µg	200%
Vitamin B9	800 µg	400%
Vitamin B12	500 µg	20,000%
Vitamin C	120 mg	150%
Vitamin D	75 µg	1500%
Vitamin E	20 mg α-ET	167%
Vitamin K1	300 µg	400%
Iron	91 mg	650%
Copper	4 mg	400%
Zinc	30 mg	300%
Iodine	150 µg	100%
Selenium	105 µg	100%
Manganese	3 mg	150%
Molybdenum	112.4 µg	225%
Chrome	160 µg	400%
Beta carotene	1.2 mg	

***** RI Recommended Intake.

**Table 3 nutrients-15-00742-t003:** Baseline demographic and clinical characteristics of people operated to SADIS-S enrolled in our study (66 patients).

Age	42 (38–50) *
Height, cm	168 (160–172) *
Weight, kg	143 (129–172) *
BMI, Kg/mq^2^	53 (48–58) *
Sex	19 male (28.78%) 47 female (71.22%)
Operative time, min	120 (106–154) *
Discharge, days	3 (2–4) *
Smoking	33 No (50%) 17 Yes, interrupted after 6 months by the surgery (25.8%) 16 Yes, interrupted within 6 months after surgery (24.2%)

***** Numbers refer to median (IQR).

**Table 4 nutrients-15-00742-t004:** Anthropometric results of people operated with SADIS-S enrolled in our study (66 patients).

BMI, Kg/m^2^	27 (23–31)
Total weight loss (TWL), kg	75 (49–100)
%TWL	50.9 (40.7–56.9)
%EWL	85.3 (72.1–96.0)

**Table 5 nutrients-15-00742-t005:** Nutritional results of people operated with SADIS-S enrolled in our study (66 patients).

		Before SADI-S		24 Months after SADI-S		
	Reference Values	Median	% Deficiency	Median	% Deficiency	*p* Value *
Hemoglobin (g/dL)	F (12.0–15.0 g/dL) M (13.0–17.0 g/dL)	13.3 (12.15–13.75)	18.18%	12.9 (11.4–13.2)	3.03%	0.012
Total Serum Protein (g/L)	(65–85 g/L)	78 (74.5–81.5)	0.00%	66 (63.5–70.1)	1.52%	0.088
Albumin (g/L)	(34–48 g/L)	41 (39.1–43.3)	1.52%	38 (34.5–42.1)	4.52%	0.078
Calcium (mg/dL)	(8.6–10.2 mg/dL)	9.6 (9.2–9.8)	0.00%	8.9 (8.6–9.2)	0.00%	0.001
Sodium (mmol/L)	(135–145 mmol/L)	140 (139–141)	0.00%	141 (139–141.0)	0.00%	0.472
Potassium (mmol/L)	(3.0–5.0 mmol/L)	3.9 (3.6–4.05)	0.00%	4.0 (2.9–4.4)	3.03%	0.241
Chloride (mmol/L)	(98–108 mmol/L)	103 (100.5–107.2)	0.00%	103 (99.0–108.5)	1.52%	0.498
HDL (mg/dL)	(>40 mg/dL)	46 (38.5–55.1)	22.72%	52 (29.0–70.5)	1.52%	0.577
LDL (mg/dL)	(<130 mg/dL)	103 (95.0–158.2)	18.18%	60 (50.6–142.6)	1.52%	0.475
HbA1c (mmol/mol)	(23.0–41.0 mmol/mol)	46 (43.1–46.5)	9.09%	30 (24.5–40.5)	0.00%	0.048
Glucose (mg/dl)	(65–110 mg/dL)	90 (84.50–106)	16.67%	81 (75.0–87.0)	0.00%	0.021
Vitamin D (ng/mL)	(31–100 ng/mL)	29.4 (16.1–38.7)	12.12%	28.8 (10.2–39.7)	31.82%	0.406
Vitamin B12 (pg/mL)	(187–883 pg/mL)	436 (373.7–1193.5)	0.00%	945 (678.0–1035.0)	1.51%	0.5
Folic acid (ng/mL)	(>4 ng/mL)	4.9 (3.15–7.92)	3.03%	6.3 (3.3–12.8)	9.09%	0.312
Parathormone (pg/mL)	(14–72 pg/mL)	63.3 (48.2–100.4)	12.12%	57.5 (34.02–111.0)	9.09%	0.931

* *p*-value refers to comparison continuous values.

**Table 6 nutrients-15-00742-t006:** Partial or total resolution of comorbidities after SADI-S (66 patients).

	Before SADI-S	After SADI-S		
Co-Morbid Condition		Partial Resolution	Total Resolution	New Diagnosis
OSAS	30	2	25	0
Hypertension	31	6	20	1
Diabetes	16	2	14	0

## Data Availability

The data presented in this study are available on request from the corresponding author. The data are not publicly available due to privacy and ethical restrictions.

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
