# Peer review of "Medium-Term Nutritional and Metabolic Outcome of Single Anastomosis Duodeno-Ileal Bypass with Sleeve Gastrectomy (SADI-S)"

_nutrients, 2023, doi:10.3390/nu15030742_

Round 1

Reviewer 1 Report

It may be interesting to distinguish the data of patients undergoing primary surgery from those undergoing revisional surgery. 

Line 16: tel: = Tel:

Line 27: BMI was 53 (48-58). kg/m2 Comorbidities = BMI was 53 (48-58) kg/m2. Comorbidities

Line 27-29: units of measure are sometimes reported before the round bracket and sometimes reported after the rounded frame; please standardize the format.

Line 104: Endocrine and Obesity Societies = better specify (world, American, European, etc.)

Line 106: What is the role of the product manufacturer in this study? The paper does not report that the firm has or has not played a role. Is there a conflict of interest between the authors and the firm? Did the patients receive the product free of charge from the bariatric centre/manufacturer, or did they purchase it online on the indication of the bariatric centre?

Line 116: Why was the study not submitted to an ethics committee? I understand.

Line 120: Table 1 ... We recommend creating the first header line (diet, kcal, foods/liquids, dietary composition)

Line 126-134: These are your laboratory analysis reference ranges more than the normal ranges.

Line 208: Table 3 (Median) has already been written in the text

Line 212: 53 kg/m2 , = 53 kg/m2,

Line 289: correct point (, = .)

Tables: Why do all tables have the first row in bold? Not being the first row, a header row.

Table 5: Table 5 shows the reference values of the laboratory of analysis as the last column; why not report the actual values of patients? 

Table 5 and Table 6 could be made more complete and transparent. If the p-value refers to the values of the two groups, the importance of the two groups should also be reported.

Line 359: correct the Author Contributions list 

Author Response

The author team would like to thank the reviewer for his/her time and on-point remarks on the methodology of our work, and for his/her comments and considerations.

  1. “Line 16: tel: = Tel:”;

 “Line 27: BMI was 53 (48-58). kg/m2 Comorbidities = BMI was 53 (48-58) kg/m2. Comorbidities”;

“Line 27-29: units of measure are sometimes reported before the round bracket and sometimes reported after the rounded frame; please standardize the format.”,

“Line 212: 53 kg/m2 , = 53 kg/m2,”;

“Line 289: correct point (, = .)”­

We apologize for the mistakes. Corrections are highlighted in the text.

  1. “Line 104: Endocrine and Obesity Societies = better specify (world, American, European, etc.)”

We agree with the reviewer’s opinion that it is important to underline which societies’ guidelines we follow in our clinical practice. For our bariatric patients we conform by the Italian Society of Bariatric Surgery and Metabolic Disorders (SICOB) and International Federation for the Surgery of Obesity and Metabolic Disorders (IFSO) guidelines. The two societies, not only define correct surgical indications and preoperative work-up but also recommend the conduct of a regular and standardized follow-up and nutrients supplementation in cases of deficiency. They also provide recommendations for multidisciplinary approach, which includes endocrinologists, dieticians, and psychologists.

  1. “Line 106: What is the role of the product manufacturer in this study? The paper does not report that the firm has or has not played a role. Is there a conflict of interest between the authors and the firm? Did the patients receive the product free of charge from the bariatric centre/manufacturer, or did they purchase it online on the indication of the bariatric centre?”

Thank you for pointing this out. It is our belief that supplementary products play an essential role in the outcomes of bariatric operations. We considered it relevant to mention the exact product used and its composition, so that other bariatric teams can compare their local supplementation protocols with ours. The manufacturing company had no role in this study and did not provide funding in any form. The patients purchased the product by themselves. If the reviewer believes that mentioning the company’s name poses a potential conflict of interest, it can be omitted and only the composition will be visible.

  1. “Line 116: Why was the study not submitted to an ethics committee? I understand.”

We understand the reviewer’s concerns over this matter. Following our initial application at the ethics committee, we were informed orally that due to the retrospective nature of the study and the de-identified database, a separate informed consent by the enrolled patients was unnecessary and at the time of the first submission we had not applied for a formal response by the ethics committee. However, patients enrolled had already signed informed consent for this study. As per the reviewer’s suggestion we since have requested and obtained a formal approval.  

  1. “ Line 120: Table 1 ... We recommend creating the first header line (diet, kcal, foods/liquids, dietary composition)”;

“Line 208: Table 3 (Median) has already been written in the text”;

“Tables: Why do all tables have the first row in bold? Not being the first row, a header row.”

According with the reviewer’s suggestions, we modified tables adding first header, removing “Median” and bold line (see Table 1 and Table 3).

  1. "Line 126-134: These are your laboratory analysis reference ranges more than the normal ranges."

We are aware of the particularly high ranges of our laboratory which certainly overestimate nutritional deficiencies. Moreover, probably, if the cut-offs were lower our study would report fewer patients with altered micronutrients with no need for subsequent treatment. We tried to specify it in our manuscript in this sentence:" While this clearly allows an early identification and treatment of these conditions, it also results in slightly higher rates than those reported by many authors, who instead use more tolerable ranges."

  1. "Table 5: Table 5 shows the reference values of the laboratory of analysis as the last column; why not report the actual values of patients?"

"Table 5 and Table 6 could be made more complete and transparent. If the p-value refers to the values of the two groups, the importance of the two groups should also be reported."

The authors greatly appreciated the reviewer’s remarks, as the corrections proposed make the tables more coherent and understandable. Tables 5 and 6 were completely reformulated. We hope that the new version is more comprehensible.

  1. "Line 359: correct the Author Contributions list"

We corrected the Author Contributions list

Reviewer 2 Report

I congratulate the authors for the good work done, the paper is well written. There are some minor elements to improve.

Here are my suggestions:

1-Line 44, please detail better the type of comorbidities related to obesity, in metabolic patients.

2-line 110. How long does the supplementation with FitForMe last? All patients take the supplement for the same period, if the answer is negative please provide a table with compliance of the supplementation with this supplement.

3-Line 132,  for folic acid, insert what kind of measure the author use, serum or RBC folic acid? Remember the serum folic acid explain the previous one week-ten days, instead RBC folic acid explains the 3 months' previous folic acid status.

4-Table 5. Why anemia was assessed only in 52 subjects instead of 66 subjects?

5-Table 6 in the text, line 229 or before, please explain what the author intends as a partial resolution of Diabetes, HBP and OSAS. The term 'partial resolution' with a lacking explanation in these 3 cases is unclear.

6-line 285, explain better what is intended for Abnormal levels. Is the level low or high or both?

Author Response

The author team would like to thank the reviewer for his/her suggestions and remarks.

1. "Line 44, please detail better the type of comorbidities related to obesity, in metabolic

patients."

The type of obesity-related comorbidity requiring metabolic surgery is considered type 2

diabetes mellitus (T2DM). In particular, hypoabsorptive surgery has been shown to have a

greater impact on T2DM; indeed, a high rate of complete remission was described. We

modified the manuscript to clarify this point.

2. “line 110. How long does the supplementation with FitForMe last? All patients take the

supplement for the same period, if the answer is negative please provide a table with

compliance of the supplementation with this supplement.”

Supplementation begins about three days after surgery and lifelong continuation is

recommended. In particular, we advised the intake of the supplement throughout the study

period.

3. “Line 132, for folic acid, insert what kind of measure the author use, serum or RBC folic

acid? Remember the serum folic acid explain the previous one week-ten days, instead RBC

folic acid explains the 3 months' previous folic acid status.”

Unfortunately, our laboratory did not allow routine RBC dosage, which would certainly

allow us a more in-depth and precise evaluation of folic acid status. In our manuscript, we

reported the values of serum folic acid which we will report, as required, for completeness.

4. “Table 5. Why anemia was assessed only in 52 subjects instead of 66 subjects?”

“Table 6 in the text, line 229 or before, please explain what the author intends as a partial

resolution of Diabetes, HBP and OSAS. The term 'partial resolution' with a lacking

explanation in these 3 cases is unclear.”

We apologize for the mistake. All patients assessed anemia, we decided to completely

reformulate the tables to make them more comprehensible.

5. “line 285, explain better what is intended for Abnormal levels. Is the level low or high or

both?”

For Global nutritional deficiency, we basically mean lower levels of micronutrients with the

exception of parathormone which in patients undergoing hypoabsorptive surgery leads to a

reduction absorption of vitamin D and increase in circulating parathyroid hormone. we have

added the changes requested to make the manuscript more understandable

Round 2

Reviewer 1 Report

Your paper is interesting, with some limitations, but still enjoyable.

Suggested corrections:

Line 30: a %TWL of 75 (49-100) kg = This refers to TWL in absolute value, expressed in kg, therefore the symbol % = TWL of 75 (49-100) kg

Line 106: I suggest you write "Bariatric Surgery Societies (SICOB, IFSO)"; if you want, you can specify in full what means SICOB and IFSO

Line 108: In my opinion, the product's name is a delicate issue because of how you wrote the text. I suggest Option 1) not to mention the product, but only its composition (clinical experts will then identify the product, as bariatric companies are not highly numerous); Option 2) state in the paper text that patients have purchased the product at their own expense (typically in research studies products are provided free of charge to patients); a state in the text of the paper that the firm had no role in this study and that you have no conflicts of interest with the firm you mention.

For example, All patients received directions to buy a vitamin and mineral bariatric supplement

Table 2. Composition of vitamin and mineral bariatric supplement

Table 3: missing (Median) in BMI, add "BMI, Kg/mq2 (Median) or delete (Median) in all other entries 

It's odd for an ethics committee to say verbally what it says. It is recommended that the actual description of how the research project has been evaluated (or not) by the ethics committee be reported in the paper text.

Table 5 is much more apparent. As suggested, I would not use it to indicate "normal values" but "reference values at our laboratory". For example, in your reading, I take for Vitamin B12 abnormal values 0%, but in the reported range (373.7-1193.5), some value outside your reference range; therefore, there is. I suggest indicating all the number of % abnormalities with two decimals after the point (X.XX format). This way that 0% will be more accurate. 

Author Response

We would like to extend our gratitude for the reviewer’s comments and suggestions.

  1. “Line 30: a %TWL of 75 (49-100) kg = This refers to TWL in absolute value, expressed in kg, therefore the symbol % = TWL of 75 (49-100) kg.”

As per the reviewer’s suggestion, we corrected the text from kg to %.

  1. “Line 106: I suggest you write "Bariatric Surgery Societies (SICOB, IFSO)"; if you want, you can specify in full what means SICOB and IFSO.”

We deeply appreciate your feedback. We followed the reviewer’s first option (abbreviations without further specifications).

  1. “Line 108: In my opinion, the product's name is a delicate issue because of how you wrote the text. I suggest Option 1) not to mention the product, but only its composition (clinical experts will then identify the product, as bariatric companies are not highly numerous); Option 2) state in the paper text that patients have purchased the product at their own expense (typically in research studies products are provided free of charge to patients); a state in the text of the paper that the firm had no role in this study and that you have no conflicts of interest with the firm you mention.

“For example, all patients received directions to buy a vitamin and mineral bariatric supplement.”

We recognize the delicacy of the topic and appreciate the reviewer’s concerns. We have removed the product’s name from the revised manuscript and made the necessary text and table adjustments (lines 108-115) and “Table 2. Composition of vitamin and mineral bariatric supplement.”

  1. “Table 3: missing (Median) in BMI, add "BMI, Kg/mq2 (Median) or delete (Median) in all other entries”

Thank you for your suggestions. We have deleted the median and IQR from the contents of the table and add it with an asterisk as a table footnote (maybe it’s more convenient this way for the reviewers/readers)

  1. “It's odd for an ethics committee to say verbally what it says. It is recommended that the actual description of how the research project has been evaluated (or not) by the ethics committee be reported in the paper text.”

Sorry for any misunderstanding. Following the reviewer’s initial remark, we applied for the ethics committee approval, explaining the urgency of the situation, and received it. So, as per the reviewer’s suggestion, we report it in the revised manuscript (line 401).

  1. Table 5 is much more apparent. As suggested, I would not use it to indicate "normal values" but "reference values at our laboratory". For example, in your reading, I take for Vitamin B12 abnormal values 0%, but in the reported range (373.7-1193.5), some value outside your reference range; therefore, there is. I suggest indicating all the number of % abnormalities with two decimals after the point (X.XX format). This way that 0% will be more accurate. written in the text”;

Thank you for your suggestions. We substituted abnormal with deficiency as it seems more appropriate and clearer. We also added the two decimals in percentages in the text and table
